# PIVNO: Particle Image Velocimetry Neural Operator

Jie Xu[*2], Xuesong Zhang[* †1], Jing Jiang[2] and Qinghua Cui[†3,4]

[1]Beijing University of Posts and Telecommunications
[2]Beijing Union University
[3]Peking University
[4]Wuhan Sports University
[1]xuesong_zhang@bupt.edu.cn
[3]cuiqinghua@hsc.pku.edu.cn

## Abstract

Particle Image Velocimetry (PIV) aims to infer underlying velocity fields from time-separated particle images, forming a PDE-constrained inverse problem governed by advection dynamics. Traditional cross-correlation methods and deep learning-based feature matching approaches often struggle with ambiguity, limited resolution, and generalization to real-world conditions. To address these challenges, we propose a PIV Neural Operator (PIVNO) framework that directly approximates the inverse mapping from paired particle images to flow fields within a function space. Leveraging a position informed Galerkin-style attention operator, PIVNO captures global flow structures while supporting resolution-adaptive inference across arbitrary subdomains. Moreover, to enhance real-world adaptability, we introduce a self-supervised fine-tuning scheme based on physical divergence constraints, enabling the model to generalize from synthetic to real experiments without requiring labeled data. Extensive evaluations demonstrate the accuracy, flexibility, and robustness of our approach across both simulated and experimental PIV datasets. Our code is at https://github.com/ZXS-Labs/PIVNO.

## 1 Introduction

Particle Image Velocimetry (PIV) is a computer vision-based metrology widely used in various scientific and engineering fields, *e.g.*, physics [1–3], materials [4–6], life sciences [7–9], engine designs [10, 11], and locomotion inspection in tissue engineering [12]. By dispersing tracer particles into the fluid under measurement, PIV employs high-frame-rate cameras (typically $10^3$ to $10^5$ fps) and high-repetition-rate laser sources (up to $10^4$ Hz) to capture particle image sequences and calculate the particle displacements, providing a discrete observation of the underlying motion field for further flow dynamic analysis. The motion of these tracer particles in an incompressible fluid can be modeled by the advection–diffusion equation [13–16]

$$\frac{\partial I}{\partial t} + \nabla \cdot (\mathbf{u}I) = D\nabla^2 I, \tag{1}$$

where $I(\vec{x}, t)$ represents the observed scalar field (e.g., particle intensity), $\mathbf{u}$ is the underlying velocity field, and $D$ is the diffusion coefficient. Under the assumption of no source terms, negligible diffusion ($D = 0$), and divergence-free flow, (1) simplifies to the pure advection equation in the operator form

$$\mathcal{B}[u](\vec{x}, t) = f(\vec{x}, t), \quad \forall \vec{x} \in X, \quad \text{where } \mathcal{B} : u \mapsto u \cdot \nabla I, \quad f(\vec{x}, t) = \frac{\partial I}{\partial t}(\vec{x}, t) \tag{2}$$

---

[*]Equal contributions.
[†]Corresponding authors.

39th Conference on Neural Information Processing Systems (NeurIPS 2025).

In the context of PIV, One is given $I(\vec{x}, t_1)$ and $I(\vec{x}, t_2)$—two noisy, discretely sampled particle images at successive time steps—and intends to infer the latent velocity field $\mathbf{u}$ that satisfies this the partial differential equation (PDE). This makes PIV fundamentally a *PDE-constrained inverse problem*, where the forward evolution is governed by physical transport dynamics, but the goal is to invert that process and recover the control variable $\mathbf{u}$ from its outcomes. Traditional PIV methods [17–21] generally rely on local cross-correlation matching techniques, but suffer from several limitations. First, the information within a single matching window is often insufficient to distinguish completely textureless particle clusters, and this issue worsens when the particle distribution is relatively uniform in the flow. Second, the size of the matching window directly affects both the spatial resolution and the accuracy of the estimated velocity field, leading to an inherent trade-off that is difficult to balance. In recent years, deep learning-based methods [22–27, 13, 28] have been proposed to represent particle image features effectively in high-dimensional latent spaces, thus reducing matching ambiguities. Moreover, the consecutive layers in deep neural networks offer larger receptive fields capable of capturing broader spatial context while preserving spatial resolution and matching accuracy. However, the inverse problem remains fundamentally ill-posed: the observed particle images are discrete, noisy, and of limited resolution, hence the numerical solution to the governing PDE is neither unique nor stable. In other words, the inverse operator of $\mathcal{B}$ in Eq.(2) may not exist.

Algorithmically, existing PIV approaches [26, 25, 27, 29, 30] solve the PDE in (2) via the construction of a cost-volume (CV) based on which the displacement array can be inferred. Nevertheless, the resolution of the CV is up to the affordable computational resources and once trained the PIV models' scalability to different measurement requirements is very limited. Inspired by the discretization invariant neural operator (NO) method, recently developed in the field of computational physics [31–35] as efficient PDE solvers, we inspect the PIV inverse problem through the lens of operator learning. Specifically, we introduce a PIV *neural operator*(PIVNO) framework that directly approximates the inverse map $\mathcal{G} : (I_{t_1}, I_{t_2}) \mapsto \mathbf{u}$, bypassing the necessity of CVs. Particularly, we devise a position informed Galerkin-type attention operator, whose approximation properties correspond to the classical Petrov–Galerkin projections[36]. This design enables our model to perceive global motion patterns while remaining numerically efficient and resolution-agnostic.

The sim2real transferring is another challenge confronting PIV models, let alone obtaining sufficient labeled data for supervision itself is expensive and often impractical in real experiments. Existing methods [37, 22–24] are primarily trained on synthetic datasets, which, despite offering perfect ground-truth flow, fail to cover all variability of real flow conditions as well as particle image qualities. Consequently, when applied to unseen flow regimes or lighting conditions, these models suffer from poor generalization. To address this, we incorporate self-supervised fine-tuning constrained by the physical incompressibility of the flow, enforcing the divergence-free conditions, to adapt the model to unlabeled real data. This strategy bridges the domain gap between synthetic and real experiments.

Finally, practical fluid experiments often demand localized flow analysis at varying resolutions [38–41]. However, the region of interest is rarely known *a priori*, and existing PIV networks are designed for uniform global inference. This limitation prevents adaptive refinement in high-shear or boundary-layer regions, restricting their utility for fine-scale investigations. In this work, we overcome this challenge by designing our operator-based architecture to support resolution-adaptive inference over arbitrary subdomains, enabling detailed flow field predictions.

In summary, our contributions can be summarized as follows:

- We propose the PIV Neural Operator, a neural operator framework that directly maps particle image pairs to flow fields, offering a function space-level approximation of fluid dynamics and improving estimation accuracy.

- We incorporate a self-supervised fine-tuning mechanism based on physical divergence constraints, enabling domain adaptation from synthetic training to real-world testing without labeled data.

- PIVNO enables resolution-adaptive flow inference over arbitrary subdomains, allowing fine-grained analysis in critical regions, which is essential for practical experimental fluid mechanics [42–44].

## 2   Related Work

**Cost-Volume-Based PIV.** Deep learning-based PIV has attracted significant attention recently. Early studies [37, 22–24] were primarily based on convolutional neural networks (CNNs) for supervised

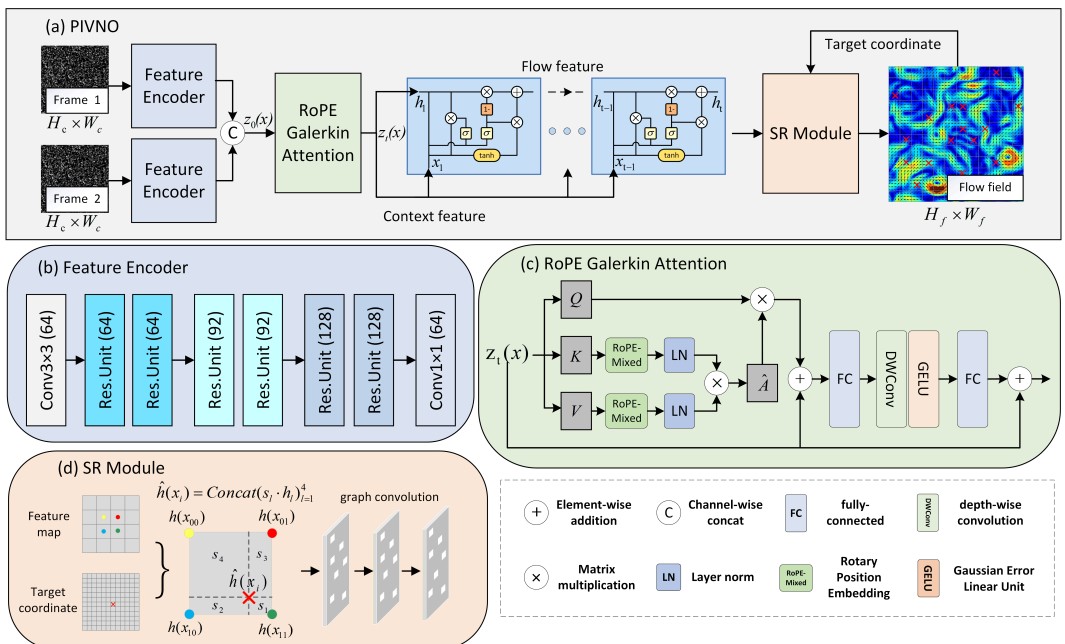

Figure 1: (a) The PIVNO framework, where each module works collaboratively to enable efficient inference from the (possible low-resolution) inputs to desired high-resolution flow fields. First, the Feature Encoder (b) extracts image features from the input pair. Next, the RoPE Galerkin Attention module (c) maps image feature functions to flow field functions via a Galerkin-style projection into the following Conv-GRU modules, where the coarse flow field features undergo iterative refinement. Finally, the SR Module (d) generates high-resolution flow field predictions through Continuous-Scale Flow Velocimetry, detailed in section 3.2.

training to estimate flow fields from images but achieved limited accuracy in flow field estimation due to their inefficiency in global motion awareness. Later work [26, 25, 27, 29, 30] proposed a cost-volume-based similarity computation to explicitly model matching relationships between image pairs, effectively improving displacement estimation accuracy. However, the construction of a four-dimensional cost volume incurs a huge computational burden and memory footprint on one hand, and more importantly freezes the possible matching accuracy and resolution determined by the grid size and dimensions of the cost volume. Therefore, cost-volume-based CNNs once trained cannot afford to solve different flow measurement problems with various accuracy requirements.

**Generalization Challenges in PIV.** Supervised learning methods depend on large labeled datasets. Due to the variability in flow conditions and particle image quality, these methods often generalize poorly to unseen flows or lighting setups. To address this, recent studies [13, 28, 45–48] have explored unsupervised optical flow algorithms for fluid flow estimation. These approaches optimize a cost function with physical constraints to estimate motion fields, offering better adaptability across diverse scenarios. However, they are highly sensitive to image quality; noise or lighting variations can significantly degrade robustness and accuracy.

## 3 Prticle Image Velocimetry Neural Operator

### 3.1 Framework of PIVNO

The goal of the proposed PIVNO is to mathematically establish a mapping $\mathcal{G} : A \rightarrow U$, where $A$ and $U$ represent the Hilbert spaces of particle image function pairs and flow fields, respectively. Specifically, we define: $A \triangleq \{a = [I_1, I_2] \in \mathbb{R}^2 | I_1, I_2 : \mathcal{X} \rightarrow \mathbb{R}, I_1(x) = I_2(x - u(x))\}$ to stress that the flow function $u(x)$ relates the input particle grayscale image pair, and $U \triangleq u : \mathcal{X} \rightarrow \mathbb{R}^2\}$ represents the function space of output flow fields. $\mathcal{X} \subset \mathbb{R}^2$ is the domain of the images. To approximate the mapping operator $\mathcal{G}$, we parameterize it with a neural network $\mathcal{G}_\theta$ and optimize the

parameters $\theta$ using supervised training. Given $n$ observations $\{a_j, u_j\}_{j=1}^n$, where $a_j$ are sampled from a distribution $\mu$ over $A$ and $u_j = \mathcal{G}(a_j)$, the training process minimizes the empirical risk:

$$\min_\theta \frac{1}{n} \sum_{j=1}^n \|u_j - \mathcal{G}_\theta(a_j)\|_U^2, \tag{3}$$

where $\|\cdot\|_U$ represents the norm in the space $U$.

As shown in fig. 1(a), PIVNO depicts a structured workflow to transform particle image pairs into accurate flow field estimates. It begins with a feature encoder that extracts localized image features as the foundational representation. The RoPE-GA module implements a Petrov–Galerkin-style projection that maps image feature functions to flow field representations. Subsequently, the Conv-GRU module iteratively optimizes the flow field features using the contextual features. Finally, the SR module reconstructs the refined features as a continuous-scale flow field representation, allowing for accurate arbitrarily scaled flow velocity estimates.

### 3.1.1 Feature Encoder

The core of flow field estimation lies in accurately matching local features; thus, extracting discriminative features is critical for matching precision. As illustrated in fig. 1(b), we design a feature encoder to extract local features from the input image pair $(I_1, I_2)$ and project them into a high-dimensional latent space. To enhance the encoder's representation capability, we integrate six residual units, each implementing a residual convolution operation defined as:

$$\mathrm{P}(I)(x) = \int_{N(x)} \kappa(x - y) I(y) \, dy + I(x), \tag{4}$$

where $\kappa(x - y)$ denotes the $3 \times 3$ convolution kernel defined over the local neighborhood $N(x)$ of position $x$, and $I(x)$ is the residual connection term. After the encoder extracts the spatial features from the image pair $I_1$ and $I_2$, the joint feature $z_0(x)$ is obtained through concatenation:

$$z_0(x) = \mathrm{Concat}(\mathrm{P}(I_1), \mathrm{P}(I_2)), \tag{5}$$

where Concat merges the two feature maps along the channel dimension; consequently, $z_0(x)$ now contains the temporal evolution of the spatial features.

### 3.1.2 RoPE Galerkin Attention

To construct a operator mapping from the image feature space to the flow field space, we propose the RoPE Galerkin Attention (RoPE-GA) module illustrated in fig. 1(c). [36] presents a general discussion on the parallelism between the finite element methods and its proposed GA module, but lacks of the positional embedding treatment for practical problems. Since the absolute positions are utmost important for particle tracking, we modify the positional encoding scheme of [49] to adapt to the Galerkin attention mechanism, leading to the enhanced GA matrix with Positional Encoding (PE) and frequency modulations:

$$\hat{A}_{j_1 j_2} = \mathrm{Re}\left[ k_{j_1}^* v_{j_2} e^{i\left( p_n^x(\theta_{j_1}^x - \theta_{j_2}^x) + p_n^y(\theta_{j_1}^y - \theta_{j_2}^y) \right)} \right], \tag{6}$$

where $p_n^x$ and $p_n^y$ denote the spatial coordinates of position $n$ along the $x$- and $y$-axes, $k_{j_1}$ and $v_{j_2}$ are the components of the key and value along channel dimensions $j_1$ and $j_2$, and $\theta_j^x$ and $\theta_j^y$ are learnable frequency parameters. See Supplementary Material Section A for detailed derivations and discussions.

The operator form of GA reads:

$$(\mathcal{A}[z])(x) \approx \sum_{j=1}^{d_z} \left( \sum_{k=1}^{d_k} \mathcal{K}_j[z](y_k) \mathcal{V}_j[z](y_k) \mathcal{Q}_j[z](x_j) \right), \tag{7}$$

where $\mathcal{K}_j[z](y_k)$ represents the evaluation of the input function $z(x)$ after the linear operator $\mathcal{K}$ at coordinate $y_k$; the index $j$ means the $j$-$th$ dimention of $\mathcal{K}(z)$. The same applies to $\mathcal{V}$ and $\mathcal{Q}$, and $\mathcal{A}$ denotes the integral kernel operator that refines $z(x)$ through the so-call basis update in [50]. However, this process is solely linear, lack of nonlinear expression capability.

In contrast, our RoPE-GA in Eq.(6) introduces frequency modulation to each channel $j$ with different learnable frequency parameter $\theta_j$. This treatment is equivalent to applying the sinusoidal activation [51] to the correlated features K and V after PE.

RoPE-GA in Eq.(6) attends to the correlation among the evaluated basis functions, $i.e.$, along the channel dimension. In order to enhance the spatial aggregation capability, we implant a $3 \times 3$ depthwise convolution between the first fully connected layer and the GELU activation function in the feed-forward network. Previous studies have shown that this operation helps capture finer spatial positional information [52–55].

### 3.1.3 GRU-based flow refinement

RoPE-GA produces coarse flow features $z_t(x)$ by mapping along the channel dimension (see Supplementary Materials Figure 1), but it does not explicitly organize spatial information; this often leads to locally inconsistent estimates. Because spatial continuity is critical for accurate motion estimation, merely stacking additional RoPE-GA layers is insufficient. We therefore introduce an iterative refinement mechanism in the spatial domain.

We adopt a convolutional GRU (Conv-GRU) for this purpose. Unlike RAFT [56], which uses image features as the context provider, our method performs recurrent refinement directly on the flow features, as illustrated in fig. 1(a). Specifically, the RoPE-GA features $z_t(x)$ serve two roles: they initialize the hidden state $h_0$ and, at every iteration, they are provided as the input feature $c_t$ to supply local neighborhood information. The Conv-GRU then updates hidden states by convolving over spatial neighborhoods, enabling iterative propagation and correction of local flow evidence. Mechanistically, the update and reset gates regulate information exchange between the current input and the previous hidden state, allowing differential nonlinear mappings across multiple local subspaces. At each iteration, the module fuses the current features with the previous estimate and adjusts both the direction and magnitude of the predicted motion through locally sensitive convolutions and gating. In this way, the Conv-GRU functions not as a temporal recurrence for sequences but as a memory-equipped, differentiable spatial transformation that progressively approximates the true flow field within local regions. In practice, we find that a small number of refinement steps suffices: five iterations achieve near-optimal performance (see Supplementary Material Section B). The operations of the Conv-GRU module can be expressed as:

$$y_t = \sigma(\text{Conv}_{3\times3}([h_{t-1}, c_t], W_z)), \tag{8}$$

$$r_t = \sigma(\text{Conv}_{3\times3}([h_{t-1}, c_t], W_r)), \tag{9}$$

$$\tilde{h}_t = \tanh(\text{Conv}_{3\times3}([r_t \odot h_{t-1}, c_t], W_h)), \tag{10}$$

$$h_t = (1 - y_t) \odot h_{t-1} + y_t \odot \tilde{h}_t. \tag{11}$$

where $\sigma$ and $\tanh$ denote the sigmoid and hyperbolic tangent activation functions, respectively; $\text{Conv}_{3\times3}$ denotes a $3 \times 3$ convolution; $W_z$, $W_r$, and $W_h$ are the learnable convolutional weights; and $\odot$ represents element-wise multiplication.

### 3.2 Continuous-Scale Flow Velocimetry

To achieve accurate super-resolution reconstruction of the flow field, the SR module (fig. 1(d)) is designed by combining random sampling, continuous interpolation, and graph convolution. The first step is adapted from SRNO [50], which enhances the scale generalizability of our model:

$$\hat{h}(x_i) = \text{Concat}(s_l \cdot h_l)_{l=1}^4 \tag{12}$$

where $x_i$ represents the target coordinate, and $l \in \{00, 01, 10, 11\}$ denotes the coordinates of the four neighboring points of $x_i$ . $s_l$ is the diagonal area of the neighboring grid point's coordinates, and $h_l$ is the feature vector of the neighboring points. The final feature vector $\hat{h}(x_i)$ is obtained by concatenating the weighted feature vectors $s_l \cdot h_l$.

The resulting feature $\tilde{h}(x_i)$ is then input into the subsequent graph convolution layers. Since the features of sampled points already incorporate positional information through the RoPE-GA module, the graph convolution [57–59] can effectively capture the spatial correlations between the randomly sampled points. By supervising on the randomly sampled points,whose results come from the graph convolution performing local information fusion, we are actually enforce PIVNO to attend to the

local correlation in fluids, enhancing the completeness and robustness of the feature representation. Finally, the graph convolution projects the high-dimensional features back into the PIV flow field solution space $u(x)$ via a $3 \times 3$ convolution:

$$u(\tilde{h})(x) = \int_{R(x)} \kappa(x - y)\tilde{h}(y)\,dy, \tag{13}$$

where $\kappa(x - y)$ represents the $3 \times 3$ graph convolution kernel defined over the set of randomly sampled points $R(x)$.

### 3.3 Self-Supervised Fine-Tuning

We further propose a self-supervised fine-tuning scheme to adapt the simulation-based pre-trained models to real experimental data. The fine-tuning strategy employs a variational optical flow method consisting of three components: a data term, a smoothing term, and a divergence regularization term. The self-supervised loss function is defined as:

$$L_P(u) = L_d(u) + \lambda_s L_s(u) + \lambda_d L_{\text{div}}(u), \tag{14}$$

where $L_d$ represents the data term, modeling the similarity of the image pairs, $L_s$ and $L_{\text{div}}$ are the spatial smoothing and divergence regularization terms respectively, and $\lambda_s$ and $\lambda_d$ are their respective weights.

**Data Term:**

$$\text{corr}(x) = \frac{\sum_{y \in N(x)} \langle I_1(y), \hat{I}_1(y) \rangle}{\sqrt{\sum_{y \in N(x)} \|I_1(y)\|^2} \cdot \sqrt{\sum_{y \in N(x)} \|\hat{I}_1(y)\|^2}} \tag{15}$$

$$L_d(I_1, I_2) = \sigma\left(1 - E_x(\text{corr})\right) \tag{16}$$

where $\hat{I}_1(x) = I_2(x - u(x))$ is the warped image compared against $I_1$, and $u(x)$ represents the flow field prediction at the spatial location $x$. $N(x)$ denotes the set of all pixel points within the sliding window. $\langle I_1(x), \hat{I}_1(x) \rangle$ is the dot product of the real image and the predicted image, and $\|I_1(x)\|$ and $\|\hat{I}_1(x)\|$ represent the magnitudes of the real image and the predicted image within the window, respectively. The function $\sigma(z) = (z^2 + \epsilon^2)^\gamma$ is the Charbonnier penalty function, used to smooth the error term $z$, where $\gamma$ controls the degree of smoothing, and $E_x$ denotes the expectation over all the positions in the images.

**Smoothing Term:**

$$L_s(u) = \sigma\left(\nabla^2 u(x)\right) \tag{17}$$

where $\nabla^2 u(x)$ is the second derivative of $u(x)$ (the Laplacian operator), which measures the local variation of the underlying flow field.

**Divergence Term:**

$$L_{\text{div}}(u) = \sigma\left(\nabla \cdot u(x)\right) \tag{18}$$

where $\nabla \cdot u(x)$ is the divergence of $u(x)$, which enforces the incompressibility constraint for fluid flow, constraint the divergence of the flow.

## 4 Experiments

The experimental evaluation comprises three synthetic datasets and three real-world PIV challenge tasks. Initially, supervised training and benchmark testing are conducted on Synthetic Datasets 1 and 2. Training samples are generated by uniformly sampling downsampling factors within the range of $1\times$ to $4\times$, allowing the model to generalize across varying output resolutions after a single training phase. Subsequently, self-supervised fine-tuning is performed on Synthetic Dataset 3 and on the three real-world PIV benchmarks to enhance cross-domain adaptability. Since ground-truth flow fields are unavailable for real-world benchmarks, direct quantitative evaluation is inherently infeasible. The selected real-world PIV tasks are chosen for their established and credible evaluation protocols, enabling reliable qualitative comparison.

In addition, ablation experiments are conducted to analyze the contribution of key modules and loss components. **Table 3** specifically presents the ablation results of the self-supervised loss terms and

Table 1: This table presents the Average Endpoint Error (AEE) on synthetic dataset 1, where the error unit is set to pixels per 100 pixels for easier comparison. All methods are evaluated using a fixed input and output resolution of $256^2$. The last four rows show the performance of our method at different input resolutions (e.g., $64^2 \times 4$ represents an input of $64^2$ with 4× upsampling to achieve the $256^2$ output). The Params column indicates the number of parameters in millions (M).

| Methods | Backstep | Cylinder | JHTDB Channel | DNS turbulence | SQG | Params. (M) |
|---|---|---|---|---|---|---|
| Farneback [60] | 8.5 | 8.3 | 14.1 | 37.8 | 33.2 | - |
| PIV-DCNN [22] | 4.9 | 7.8 | 11.7 | 33.4 | 47.9 | 8.40 |
| PIV-LiteFlowNet [23] | 5.6 | 8.3 | 10.4 | 19.6 | 20.0 | 6.25 |
| PIV-LiteFlowNet-en [23] | 3.3 | 4.9 | 7.5 | 12.2 | 12.6 | 5.59 |
| UnPwcNet-PIV [13] | 8.2 | 7.1 | 13.4 | 21.5 | 25.2 | 9.37 |
| UnLiteFlowNet-PIV [13] | 9.4 | 6.9 | 8.4 | 15.0 | 17.3 | 5.38 |
| OFVNetS [61] | 15.0 | 1.6 | 25.0 | 8.3 | 22.3 | - |
| OFVNetS-HS [61] | 13.7 | 4.7 | 32.7 | 7.0 | 18.9 | - |
| PIV-RAFT [25] | 1.6 | 1.4 | 13.7 | 9.3 | 11.7 | 5.31 |
| ARaft-FlowNet [29] | 3.1 | 2.0 | 8.3 | 9.6 | 9.8 | - |
| **PIVNO**($256^2 \times 1$) | 1.9 | **0.8** | **1.7** | **3.5** | **2.5** | |
| **PIVNO**($128^2 \times 2$) | **0.9** | 1.1 | 2.8 | 4.7 | 4.1 | **2.52** |
| **PIVNO**($64^2 \times 2$) | 1.0 | 1.4 | 2.9 | 5.0 | 4.2 | |
| **PIVNO**($64^2 \times 4$) | 2.7 | 3.1 | 11.3 | 18.0 | 18.0 | |

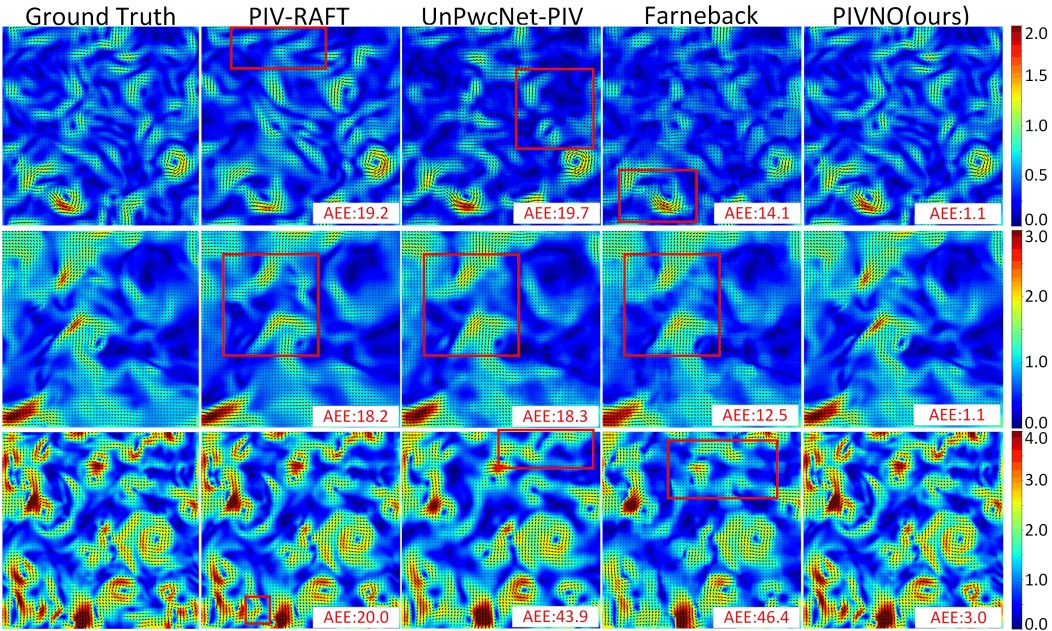

Figure 2: Visualization of three flow fields: DNS turbulence (first row), JHTDB channel flow (second row), and SQG (third row). The rectangles indicate over-smoothing or even failure artifacts existing in the outputs of comparative methods, while PIVNO faithfully recovers the flow.

simultaneously summarizes the overall performance of the self-supervised strategy on both synthetic and real-world datasets. It is therefore presented separately to highlight its central role in validating the effectiveness of the proposed self-supervised strategy. **Section 4.3** further investigates the influence of architectural components. Beyond the main results, we also conduct extended analyses, including statistical robustness evaluation (maximum error and standard deviation), cross-domain fine-tuning generalization experiments, and zero-shot resolution generalization studies. All of these additional results are provided in the Supplementary Material for completeness. Comprehensive details of the network architecture, dataset configurations, implementation methods, and hyperparameter settings are also included therein.

Table 2: The AEE on synthetic dataset 2. Representative baselines include the unsupervised Un-LiteFlowNet and the self-supervised PIV-RAFT.

| Methods | Backstep | Cylinder | JHTDB Channel | DNS turbulence | SQG |
|---|---|---|---|---|---|
| UnLiteFlowNet-PIV [24] | 12.3 | 7.9 | 14.5 | 22.5 | 21.6 |
| UnLiteFlowNet32-PIV [24] | 40.9 | 65.9 | 41.9 | 44.3 | 40.1 |
| PIV-RAFT [25] | 6.4 | 5.2 | 22.8 | 19.7 | 24.9 |
| **PIVNO**($256^2 \times 1$) | 4.5 | **3.4** | **4.7** | **4.4** | **6.6** |
| **PIVNO**($128^2 \times 2$) | 4.1 | 3.6 | 8.8 | 9.6 | 13.2 |
| **PIVNO**($64^2 \times 2$) | **3.4** | 5.1 | 8.8 | 10.2 | 13.9 |
| **PIVNO**($64^4 \times 4$) | 4.7 | 6.3 | 19.6 | 21.7 | 27.9 |

Table 3: Impact of loss term combinations. The table shows AEE and divergence results on SPID (sim) and solid body rotation (real).

| $L_d$ | $L_s$ | $L_{div}$ | SPID | | Solid Body Rotation Flow | |
|---|---|---|---|---|---|---|
| | | | AEE | div. | AEE | div. |
| $\times$ | $\times$ | $\times$ | 1.62 | 0.23 | 0.64 | 458 |
| $\checkmark$ | $\times$ | $\times$ | 5.00 | -8036.29 | 0.24 | -2676 |
| $\times$ | $\checkmark$ | $\times$ | 4.20 | -12.46 | 5.43 | 88 |
| $\times$ | $\times$ | $\checkmark$ | 3.72 | **-0.09** | 5.45 | **7.08** |
| $\checkmark$ | $\checkmark$ | $\times$ | 1.10 | -6639.26 | 0.23 | -2072.67 |
| $\checkmark$ | $\checkmark$ | $\checkmark$ | **0.60** | -13.00 | **0.17** | -1424.37 |

## 4.1 Evaluation on Synthetic Datasets

**Synthetic Dataset 1:** This PIV dataset [23] contains five classic flow field cases commonly used for training and benchmarking PIV algorithms. As shown in table 1, the proposed PIVNO model consistently outperforms state-of-the-art (SOTA) methods across all evaluation metrics at a resolution of $256^2$. Its advantage is especially notable in complex flow regimes such as DNS turbulence, JHTDB channel flow, and SQG sea surface flow. A key strength of PIVNO lies in its training strategy: the model is trained once on samples uniformly downsampled by factors from $1\times$ to $4\times$, enabling robust generalization across multiple upsampling scales during inference. Notably, even with low-resolution inputs (e.g., $64^2$), PIVNO produces high-resolution outputs with accuracy comparable to models using full-resolution inputs. Furthermore, when the output resolution is twice that of the input, PIVNO still achieves superior velocity field estimation, demonstrating strong robustness and adaptability in low-resolution settings.

Additionally, fig. 2 presents visual comparisons. The first row displays DNS turbulence, featuring rich small-scale vortical structures and multi-scale turbulence interactions, demonstrating high complexity and dynamic characteristics. Comparatively, only PIVNO effectively captures both local features and global relationships. The second row illustrates JHTDB channel flow, marked by shear effects, stratified velocity gradients, boundary confinement, and turbulent transition behavior. Remarkably, PIVNO accurately captures these boundary layer features. The third row shows SQG sea surface flow, including nonlinear interactions between large-scale background fields and small-scale disturbances, mainly exhibiting two-dimensional quasi-geostrophic characteristics. In summary, PIVNO handles these complex dynamics with precision.

**Synthetic Dataset 2:** To evaluate the model's performance under large displacement and high noise conditions, we used a synthetic dataset from [25], which simulates large particle displacements, low particle density, and significant noise—providing a suitable testbed for assessing the robustness of PIV algorithms. As shown in table 2, models like UnLiteFlowNet-PIV, which rely on photometric loss, suffer in high-noise settings due to disrupted motion feature extraction. PIV-RAFT also struggles with large displacements due to limitations in its local correlation-based approach. In contrast, PIVNO maintains stable prediction performance even under such challenging conditions.

**Synthetic Dataset 3:** To evaluate the impact of the three loss terms ($L_d$, $L_s$, and $L_{div}$) on flow field estimation, we conducted ablation experiments using the SPID dataset [62], which simulates real-world conditions such as noise, particle distribution, and out-of-plane motion. We progressively removed each loss term and evaluated the changes in AEE and divergence metrics. As shown in table 3, we observed that using any single loss function in isolation resulted in a significant performance degradation, with increased AEE and anomalous divergence values, indicating that a single loss term cannot effectively constrain the model's fine-tuning. In contrast, the combination of all three loss terms yielded substantial performance gains, demonstrating their synergistic effect in optimizing flow field estimation. The consistent performance deterioration when omitting any loss function further validates the effectiveness of this fine-tuning strategy.

## 4.2 Generalizability on Real PIV Challenges

**Solid Body Rotation Flow:** To evaluate the generalization capability of the fine-tuning strategy in real-world scenarios, we selected the solid body rotation flow [63] as a classical benchmark due to its theoretical clarity, uniform vorticity, and strict adherence to solid body rotation. table 3 quantifies the impact of fine-tuning, showing that combining all three loss terms ($L_d$, $L_s$, and $L_{div}$) significantly

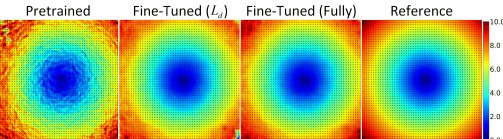

Figure 3: Comparison of the flow estimation results in the solid body rotation test.

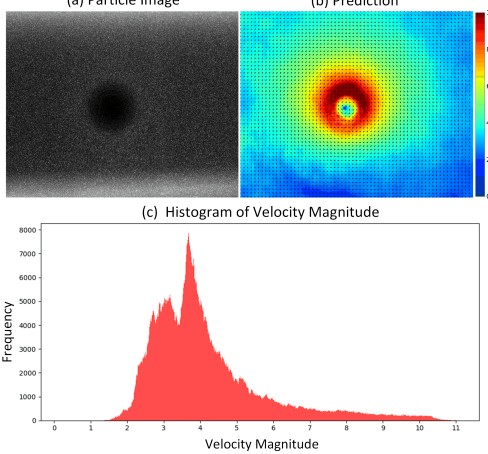

Figure 4: (a) Strong vortex particle image; (b) Estimated flow field via proposed method; (c) Velocity magnitude histogram. PIVNO accurately estimates flow and avoids "peak locking".

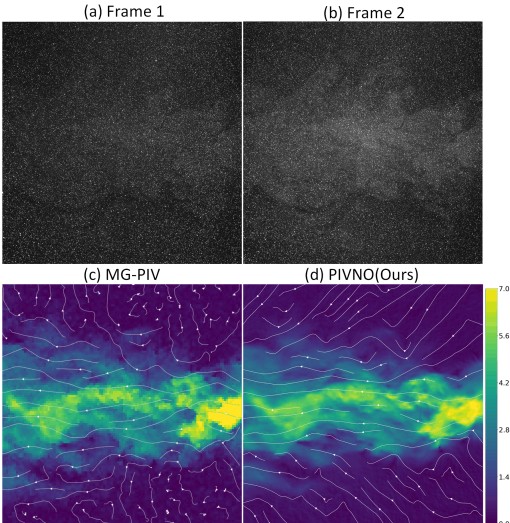

Figure 5: Comparison of turbulent round jet velocity fields: (a) and (b) are the original particle images with significant illumination variation; (c) is the fluid vector field estimated using the multigrid PIV method, appearing blurry and discontinuous; (d) is the fluid vector field estimated using PIVNO, exhibiting fair structures and continuity. The white lines indicate significant motion trajectories.

reduces AEE and improves divergence, reinforcing the model's robustness. fig. 3 illustrates this effect by comparing predictions with theoretical values, where the pre-trained model fails to capture rotational characteristics, showing flow discontinuities near the rotation center. While using only $L_d$ mitigates some errors, it still yields boundary anomalies and lacks smoothness. In contrast, the full fine-tuning strategy enables precise particle displacement prediction, closely aligning with the theoretical solution and ensuring high-quality flow estimation across both boundary and rotation center regions. These findings highlight the critical role of the three-loss synergy in enhancing model generalization and robustness for flow prediction.

**Strong Vortex:** We use PIVNO to process the vortex flow field images recorded by the German Aerospace Center in the DNW-LLF large wind tunnel [64]. The motion field contains complex characteristics such as high velocity gradients, particle density loss, size variations and small particles, making it ideal for testing the robustness of our method. Experimental results show that our approach accurately reconstructs intricate velocity distributions even under low particle density and steep velocity gradients, as illustrated in fig. 4(b). In the experimental images, particle sizes are less than two pixels, causing grayscale distributions to suffer from pixelation effects, which hinder precise subpixel-scale localization. During displacement measurement, this blurring effect biases measured values toward integer pixel positions rather than forming a continuous distribution, leading to the so-called peak-locking effect [65] and reducing accuracy. Our method effectively mitigates this issue.

**Turbulent Jet:** We evaluated the turbulent round jet dataset from Delft University of Technology [66], focusing on high-gradient regions and flow continuity. As shown in fig. 5(a) and (b), significant lighting changes between the two frames challenged flow field estimation. To obtain the velocity field, the dataset provider uses a multi-grid PIV method (fig. 5(c)). However, due to resolution limits of the grid-based approach, it struggles to resolve fine-scale flow structures, causing visible blurring artifacts. In the experiment, fluid is injected from middle-right toward middle-left, while the upper and lower sections are expected to flow inward due to pressure differences. Nevertheless, the multi-grid PIV method shows irregular and discontinuous flow in these regions, failing to preserve

Table 4: Ablation comparisons on Synthetic Dataset 1.

| RoPE-Mixed | DWConv | GA | GRU | GCN | Uniform | Backstep | Cylinder | JHTDB Channel | DNS turbulence | SQG |
|---|---|---|---|---|---|---|---|---|---|---|
| | | | × | | 10.73 | 15.55 | 4.12 | 8.95 | 20.01 | 14.96 |
| | | | | × | 4.04 | 3.07 | **0.78** | 1.88 | 3.94 | 2.90 |
| × | × | × | | | 6.68 | 2.30 | 0.95 | 1.93 | 3.85 | 3.06 |
| × | × | | | | 3.31 | 2.43 | 1.08 | 1.73 | 3.57 | 2.61 |
| × | | | | | 4.03 | 2.03 | 0.84 | 1.84 | 3.67 | 2.73 |
| | × | | | | 3.40 | 2.56 | 0.87 | 1.71 | 3.50 | 2.57 |
| | | | | | **3.26** | **1.91** | 0.79 | **1.68** | **3.47** | **2.54** |

expected motion coherence. In contrast, our method (fig. 5(d)) captures the central flow trend in both upper and lower regions more effectively, yielding a more structured and continuous velocity field.

### 4.3 Ablation Studies

We conducted ablation studies on Synthetic Dataset 1 to assess the importance of each module in PIVNO. As shown in Table 4, removing any single component degrades performance. Notably, the GA and GRU modules have the most significant impact when being removed, indicating their essential roles in the overall architecture. Other components such as RoPE-Mixed and DWConv also contribute consistently. These results validate the necessity of the full model design. More ablation experiments can be found in the supplementary material (Case B).

## 5   Conclusion

We propose PIVNO, a neural operator framework that formulates PIV as a PDE-constrained inverse problem. By leveraging a Galerkin-style attention mechanism and a self-supervised fine-tuning scheme grounded in physical constraints, PIVNO achieves accurate and resolution-adaptive flow estimation across synthetic and real-world datasets. The framework demonstrates strong generalization and robustness capabilities, highlighting its potential for high-precision PIV applications.

**Limitations.** 2D particle velocimetry cannot fully capture the true 3D nature of fluid dynamics. Future work will focus on extending the framework to 3D flow field estimation. Additionally, the current feature extraction module lacks multi-scale feature fusion, which should be taken into consideration when processing high-resolution PIV data.

## Acknowledgments and Disclosure of Funding

This work was supported in part by the National Natural Science Foundation of China under Grants 61871055, 62025102, and 82427801.

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
