# OpenReview forum: "PIVNO: Particle Image Velocimetry Neural Operator"
_NeurIPS.cc/2025/Conference — NeurIPS 2025 poster_

### Official Review · Reviewer_jXTM · 2025-06-29

**Clarity:** 3
**Significance:** 3
**Originality:** 2
**Rating:** 5
**Confidence:** 4

**Summary:**

The paper presents PIVNO, a neural operator approach for Particle Image Velocimetry that directly maps pairs of particle images to velocity fields. It combines Galerkin-style attention, Conv-GRU refinement, and a self-supervised fine-tuning procedure grounded in physical constraints. Evaluated on both synthetic and experimental data, PIVNO achieves accurate and resolution-adaptive flow estimation.

**Questions:**

- Could you elaborate on the role and formulation of the smoothing term in the fine-tuning loss function?

- Is it necessary to use graph convolution layer to achieve accurate super-resolution? Have you considered other approaches?

- What are the implications for the model's performance when the inverse problem is ill-posed or lacks a unique solution?

- Could the authors clarify the motivation behind selecting GRU-based refinement and RoPE Galerkin Attention for their architecture, and explain how these choices compare to potential alternatives in terms of performance or inductive bias?

**Ethical Concerns:**

["NO or VERY MINOR ethics concerns only"]

**Final Justification:**

The authors have addressed my concerns, and I am confident that this work should be accepted, even though, to the best of my knowledge, this type of work is not typically submitted to NeurIPS.

**Limitations:**

The authors clearly explained the limitations of their approach.

**Paper Formatting Concerns:**

/

**Quality:**

3

**Strengths And Weaknesses:**

Strengths

- The paper evaluates the model on a wide range of benchmarks, including both synthetic and real-world datasets.
- The model's performance is compared against numerous baselines, including several state-of-the-art methods such as PIV-RAFT and PIV-LiteFlowNet.
- Although the model involves a complex engineering design, the ablation studies convincingly demonstrate that each component plays a necessary and justified role in achieving the overall performance.
- The proposed self-supervised fine-tuning strategy, grounded in physical constraints like incompressibility, is a practical and effective solution for adapting the model to real-world data without requiring labeled ground truth.

Weaknesses

- This is a very interesting and well-executed study on Particle Image Velocimetry. However, despite the paper's strengths, its scope is limited to a specific class of PDE-constrained inverse problems and does not readily generalize to broader categories of such problems.

- The paper does not include a comparison between PIVNO and well-established neural operator baselines, such as FNO [1] and CNO [2], or some generic neural operators designed for inverse problems [3],  which could be readily adapted to this class of inverse problems.

- The rationale behind the authors’ choice of GRU-based flow refinement and RoPE Galerkin Attention is not clearly explained and would benefit from further elaboration in the paper.

----

[1] Li, Z., Kovachki, N., Azizzadenesheli, K., Liu, B., Bhattacharya, K., Stuart, A., \& Anandkumar, A. (2020). Fourier neural operator for parametric partial differential equations. arXiv preprint arXiv:2010.08895.

[2] Raonic, B., Molinaro, R., De Ryck, T., Rohner, T., Bartolucci, F., Alaifari, R., ... \& de Bézenac, E. (2023). Convolutional neural operators for robust and accurate learning of PDEs. Advances in Neural Information Processing Systems, 36, 77187-77200.

[3] Molinaro, R., Yang, Y., Engquist, B., \& Mishra, S. (2023). Neural inverse operators for solving PDE inverse problems. arXiv preprint arXiv:2301.11167.

---

> ### Author Rebuttal · Authors · 2025-07-31
>
> Thank you very much for your valuable feedback and suggestions! We greatly appreciate your comments and treat them carefully. Below, we provide a brief discussion of the concerns raised in the review:
>
> 1. GRU-Based Flow Field Refinement and RoPE Galerkin Attention Mechanism
>
> We sincerely appreciate your specific attention to the RoPE-GA mechanism and the GRU-based flow field refinement module. Regarding the design motivation for the RoPE-GA mechanism, we have provided a detailed explanation of its principles in Section 3.1.2 of the main text and Supplementary Materials A1. The discussion of comparisons with established neural operator benchmarks has been addressed in the response to the next question. As for the design motivation behind the GRU, since several reviewers have shown interest in this aspect, we will provide a more detailed response here.
>
> Before applying Conv-GRU, we obtained flow features through the RoPE-GA module (see Supplementary Materials Figure 1). However, the RoPE-GA module only maps function spaces along the channel dimension but does not aggregate or organize the spatial information. This results in flow features that are often coarse, discontinuous, and contain local estimation errors, whereas spatial continuity is critical for accurate flow estimation. Therefore, we need a mechanism that can progressively optimize flow features in the spatial domain, which is why we did not simply stack more RoPE-GA layers.
>
> In motion estimation fields such as PIV, optical flow, and disparity estimation, the Conv-GRU module has become a mainstream structure for iterative flow refinement, with methods such as RAFT and FoundationStereo adopting similar structures. The effectiveness of this approach has been widely validated, and we did not elaborate on the structural motivation in the main text. Our ablation experiments also confirmed the key role of this module in the final performance. However, we fully recognize your rigorous concern about this issue and are happy to further explain our understanding of its mechanism.
>
> Unlike conventional convolution operations, the hidden state update in Conv-GRU is determined by multiple gated channels (such as the update gate 𝑧, reset gate 𝑟, and candidate state). Convolution essentially projects the input features into a subspace spanned by a set of convolutional kernel functions, and the optimization of these kernels allows for the best possible hidden representation. Comparatively, the convolutions in Conv-GRU operate separately on the current input features and on the previous hidden state, enabling differential nonlinear mappings across multiple subspaces during the update process. This structure allows the model to understand local motion trends from multiple perspectives and scales, and to iteratively correct the flow states of local regions at each iteration. Additionally, the nonlinear activation functions (e.g., sigmoid, tanh) introduced by the gating mechanism enhance the model's expressiveness, enabling it to adaptively regulate the retention, update, and fusion of information based on the current estimation state, thus overcoming the limitations of single-layer convolutions in modeling capacity.
>
> Throughout the iterative process, Conv-GRU acts as a differentiable spatial transformation module with memory. In each iteration, it fuses the current input features with the previous estimation result (i.e., the hidden state) and continuously adjusts the direction and magnitude of the flow field estimate through locally sensitive convolutional mappings and gating mechanisms. This structure endows the model with the ability to progressively approximate the true flow field within local spatial ranges. Therefore, Conv-GRU in PIVNO is no longer a recursive structure used for handling time sequences but rather an iterative optimization unit with local geometric modeling capabilities that supports differentiated state updates. Its synergistic effect of spatially varying linear combinations and nonlinear gating mechanisms enables the correction and refinement of the local flow field.
>
> 2. Inverse Problems and Neural Operators
>
> We appreciate the reviewers' questions regarding the applicability of inverse problem categories and comparisons with other neural operator methods. We provide a unified response here and further clarify the fundamental differences between the problem type addressed in this work and existing neural operator approaches.
>
> The problem we investigate belongs to a special class of PDE-constrained inverse problems: recovering the underlying fluid motion field from particle image pairs. This problem is not only constrained by physical laws but also influenced by the imaging mechanism, making it essentially a visual inverse problem that combines computer vision and physical modeling, rather than a traditional PDE numerical solving task based on function samples (such as field value pairs). Therefore, there are significant differences when compared to methods like FNO [1], CNO [2], or neural operator frameworks designed for general inverse problems [3].
>
> Specifically, these neural operators typically map between function spaces based on structured grids or a small number of scattered sample points, assuming well-defined problems with finite bandwidth, often suppressing high-frequency information during modeling (such as directly filtering out high-frequency modes in FNO). In contrast, the input for PIV consists of actual image pairs, which involve complex issues such as repeated patterns, occlusions, and uneven particle distributions. These issues manifest in the image space as significant local ambiguities and non-uniqueness, making the problem highly ill-posed. As a result, the solution is not unique. Therefore, PIV requires not only modeling the global mapping structure but also recovering fine-grained local high-frequency motion features, which is difficult to achieve directly using traditional neural operator paradigms.
>
> For this reason, in PIVNO, we combine an image encoder, a RoPE-GA-based neural operator representation, and a Conv-GRU-based spatial refinement module. Notably, the introduction of Conv-GRU, whose mechanism has been widely validated in computer vision inverse problems such as optical flow and disparity estimation, allows for local state memory and nonlinear optimization within the image space through gating mechanisms, a feature not applicable to traditional PDE numerical problems.
>
> Thus, our decision not to directly compare with the above methods is not for dodge but stems from the fundamental differences in problem paradigms, input modalities, and modeling objectives.
>
> Additionally, regarding the question, "What are the implications for the model's performance when the inverse problem is ill-posed or lacks a unique solution?"  we believe that PIV itself is a typical ill-posed inverse problem. Our experimental results under real-world conditions, including significant occlusions, sparse textures, and high shear deformations, already demonstrate the robustness and generalization ability of our framework in dealing with such ill-posed problems.
>
> 3. Graph Convolution
>
> The introduction of the graph convolution layer primarily stems from the cross-resolution training strategy used in the SR Module. This strategy randomly samples the same number of pixels for training from image pairs at different resolutions, enhancing the model's generalization ability under scale variations (see Supplementary Materials B3 for details). However, this sampling method disrupts the regular, evenly spaced arrangement of pixels, making it difficult for traditional convolutions (such as standard 2D convolutions) to directly operate on these non-structured sampling points. Since capturing local features is critical for continuous spatial mapping, graph convolution naturally handles non-uniformly sampled inputs and aggregates local information through adjacency structures, enabling effective continuous mapping and feature fusion in space. Therefore, we believe that graph convolution is a reasonable choice for the task scenario at hand.
>
> To verify its necessity, we replaced the graph convolution with a 1×1 convolution in an ablation experiment (the "GCN×" setting in Table 4). The results showed a significant performance drop, further demonstrating the unique advantages of graph convolution in handling irregular sampling points and capturing local geometric features.
>
> 4.Smoothness Term
>
> For the fluid motion field $ u(\mathbf{x}) \in \mathbb{R}^{C \times H \times W} $, the smoothness term in the loss function is defined as follows:
>
> $L_{\text{smooth}} = \sum_{c=1}^{C} \sum_{i=2}^{H-1} \sum_{j=2}^{W-1} \frac{1}{(H-2)(W-2)} \left( \nabla^2 u_{c,i,j} \right)^2$
>
> where $ \nabla^2 u_{c,i,j} $ denotes the Laplacian of $ u $ at position $ (i,j) $. Specifically, the Laplacian is defined as:
>
> $\nabla^2 u_{c,i,j} = u_{c,i+1,j} + u_{c,i-1,j} + u_{c,i,j+1} + u_{c,i,j-1} - 4u_{c,i,j}$
>
> The smoothing term is designed based on the spatial continuity assumption and regularization theory. The spatial continuity assumption posits that in the physical world, unless there are object boundaries or occlusions, the fluid velocity field is typically continuous and smoothly varying in space, with abrupt changes being rare. The smoothing term aims to explicitly express and reinforce this prior knowledge, encouraging the estimated flow field to remain smooth and consistent spatially. From a regularization perspective, the particle image velocimetry problem is inherently an ill-posed inverse problem, and relying solely on data terms (such as brightness consistency) often struggles to produce a stable and reasonable solution. Introducing the smoothing term as a regularization term effectively constrains the spatial fluctuations of the solution, prevents overfitting to noise, and improves the stability and robustness of the model's estimation.

---

> > ### Comment · Reviewer_jXTM · 2025-08-04
> >
> > I appreciate the authors’ response and the clarifications provided. I am increasing my score to 5.

---

> > > ### Author Response · Authors · 2025-08-05
> > > **Thank you**
> > >
> > > Thank you so much for taking the time to read our paper and the responses. We are grateful for the improved score.

---

### Official Review · Reviewer_PrER · 2025-06-30

**Clarity:** 3
**Significance:** 3
**Originality:** 3
**Rating:** 4
**Confidence:** 5

**Summary:**

This paper addresses the problem of Particle Image Velocimetry (PIV), which aims to infer underlying velocity fields from time-separated particle images. The core idea is to directly learn the inverse mapping from paired particle images to the corresponding flow fields within a function space. The proposed framework utilizes a position-informed Galerkin-style attention operator (RoPE-GA) to capture global flow structures while enabling resolution-adaptive inference. Furthermore, to improve generalization to real-world conditions, the paper introduces a self-supervised fine-tuning scheme based on physical divergence constraints, which allows the model to adapt from synthetic to real experimental data without requiring labels.

**Questions:**

1/ Regarding the GRU refinement module, beyond its function of iterative optimization, is there a deeper motivation for choosing the Conv-GRU structure specifically? For example, what advantages do the recurrent and gating mechanisms of a GRU offer for the task of flow field refinement compared to simply stacking more RoPE-GA layers or using other types of convolutional networks for feature fusion?

2/ A key claimed advantage of the framework is its resolution adaptability. The experiments in Table 1 demonstrate robustness to low-resolution inputs. To more rigorously validate its properties as a true neural operator, have you considered performing a "zero-shot resolution generalization" test? For instance, training the model on a single resolution (e.g., 128x128) and then evaluating its performance on a variety of unseen output resolutions (e.g., 192x192, 256x256, etc.). This would provide stronger evidence of the model's generalization capabilities.

3/ Your ablation in Table 3 shows that using only the divergence loss ($L_{div}$) leads to poor performance. This is an interesting result, especially since other works, such as [1], have successfully used a zero-divergence loss. This suggests the effectiveness of this physical prior is highly context-dependent. Could you provide further discussion on this? For example, under what conditions is the divergence loss a strong regularizer on its own, and when does it require a strong data term to be effective?

[1] Dual-frame Fluid Motion Estimation with Test-time Optimization and Zero-divergence Loss

**Ethical Concerns:**

["NO or VERY MINOR ethics concerns only"]

**Final Justification:**

Please see my cmt below.

**Limitations:**

Please see the section of weaknesses and questions.

**Quality:**

3

**Strengths And Weaknesses:**

Strengths

1/ Novel and Insightful Problem Formulation: Framing the PIV problem as an operator learning task is a strong and modern approach. This conceptual shift from simple feature matching to learning the underlying physical mapping is a significant contribution that is more aligned with the physical nature of the problem and has the potential to overcome the inherent trade-offs of traditional methods.

2/ Innovative and Well-Motivated Architecture: The proposed RoPE-GA module is a key innovation with clear motivation. The authors correctly identify that standard Galerkin Attention lacks the absolute positional information crucial for PIV and creatively integrate Rotary Position Embeddings (RoPE) to address this gap. This results in a well-designed and well-justified architecture.

Weaknesses

1/ Insufficient Motivation for Some Network Components: While the motivation for the core RoPE-GA module is very clear, the justification for other architectural choices could be stronger. For example, the paper states that a Conv-GRU module is used for iterative flow refinement, but does not provide a deep justification for why the GRU architecture is specifically optimal for this task compared to other iterative methods or feature-fusion strategies.

2/ Incomplete Ablation of Self-Supervised Loss Terms: In Table 3, the paper shows that the full three-term loss function performs best. However, this ablation does not fully isolate the individual contributions of each term, particularly the smoothing term (Ls) and the divergence term (Ldiv). The lack of results for combinations like (Ld + Ls) or (Ld + Ldiv) makes it difficult to clearly understand the specific benefit that the physics-based constraint adds over a traditional smoothing regularizer.

3/ Missing Inference Speed Comparison: The paper emphasizes that its operator-based approach avoids the need for computationally expensive cost volumes, implying a potential advantage in inference speed. However, the experimental section lacks a direct comparison of inference times between PIVNO and key baselines like PIV-RAFT. Such a comparison would be essential for a complete assessment of the method's practical advantages.

---

> ### Author Rebuttal · Authors · 2025-07-30
>
> Thank you very much for your valuable feedback and suggestions! We greatly appreciate your comments and treat them carefully. For each issue raised, we will make modifications or clarifications as needed. Below, we provide a brief discussion of the concerns raised in the review:
>
> 1. On the Conv-GRU Structure
>
> We fully understand your concerns regarding the Conv-GRU structure, and we would like to briefly explain the deeper motivation behind our choice of the Conv-GRU iterative architecture.
>
> Before applying Conv-GRU, we obtained flow features through the RoPE-GA module (see Supplementary Materials Figure 1). However, the RoPE-GA module only maps function spaces along the channel dimension but does not aggregate or organize the spatial information. This results in flow features that are often coarse, discontinuous, and contain local estimation errors, whereas spatial continuity is critical for accurate flow estimation. Therefore, we need a mechanism that can progressively optimize flow features in the spatial domain, which is why we did not simply stack more RoPE-GA layers.
>
> In motion estimation fields such as PIV, optical flow, and disparity estimation, the Conv-GRU module has become a mainstream structure for iterative flow refinement, with methods such as RAFT [2] and FoundationStereo [3] adopting similar structures. The effectiveness of this approach has been widely validated, and we did not elaborate on the structural motivation in the main text. Our ablation experiments also confirmed the key role of this module in the final performance. However, we fully recognize your rigorous concern about this issue and are happy to further explain our understanding of its mechanism.
>
> Unlike conventional convolution operations, the hidden state update in Conv-GRU is determined by multiple gated channels (such as the update gate 𝑧, reset gate 𝑟, and candidate state). Convolution essentially projects the input features into a subspace spanned by a set of convolutional kernel functions, and the optimization of these kernels allows for the best possible hidden representation. Comparatively, the convolutions in Conv-GRU operate separately on the current input features and on  the previous hidden state, enabling differential nonlinear mappings across multiple subspaces during the update process. This structure allows the model to understand local motion trends from multiple perspectives and scales, and to iteratively correct the flow states of local regions at each iteration. Additionally, the nonlinear activation functions (e.g., sigmoid, tanh) introduced by the gating mechanism enhance the model's expressiveness, enabling it to adaptively regulate the retention, update, and fusion of information based on the current estimation state, thus overcoming the limitations of single-layer convolutions in modeling capacity.
>
> Throughout the iterative process, Conv-GRU acts as a differentiable spatial transformation module with memory. In each iteration, it fuses the current input features with the previous estimation result (i.e., the hidden state) and continuously adjusts the direction and magnitude of the flow field estimate through locally sensitive convolutional mappings and gating mechanisms. This structure endows the model with the ability to progressively approximate the true flow field within local spatial ranges. Therefore, Conv-GRU in PIVNO is no longer a recursive structure used for handling time sequences but rather an iterative optimization unit with local geometric modeling capabilities that supports differentiated state updates. Its synergistic effect of spatially varying linear combinations and nonlinear gating mechanisms enables the correction and refinement of the local flow field.
>
> 2. Self-Supervised Fine-Tuning
>
> Thank you for your attention and inquiries regarding the ablation experiments on the self-supervised loss terms in Table 3. We provide a unified response addressing both the theoretical basis and experimental design.
>
> First, regarding your observation that “using only the divergence loss term $L_{\text{div}}$ leads to significant performance degradation,” we explain that this behavior is closely related to the ill-posed inverse problem nature of PIV. PIV maps observed image pairs to fluid motion fields, with inputs containing uncertainties such as repeated patterns, occlusions, and varying lighting conditions, making the inverse problem highly ill-conditioned and solutions non-unique. Under such conditions, a single regularization term is insufficient to constrain the solution space, easily causing the model to converge to physically unreasonable spurious solutions, thus losing estimation effectiveness.
>
> In contrast, the work you mentioned [1] is based on a PTV framework where inputs are explicitly detected particle spatial coordinates, providing strong, unique observational support that substantially reduces the ill-posedness of the inverse problem. It is important to note that [1] also employs a combined loss of data consistency, smoothness, and divergence constraints during training. The “divergence-only” setting you referred to occurs only during testing, where the backbone network parameters are frozen and local adjustments are made via a residual module. Under such conditions, the “divergence-driven” mechanism does not independently constrain the backbone network, and thus cannot be directly compared to our independent loss configuration during training.
>
> Second, regarding your concern that the ablation experiments do not fully reveal the individual contributions of each loss term, we acknowledge that not all possible loss combinations were presented in the main paper, but with primary focus on the core experimental questions. Our experimental design emphasizes two aspects: (1) verifying whether a single loss term is sufficient to avoid invalid solutions in PIV scenarios; and (2) assessing whether adding the physics-based divergence constraint to the traditional variational optical flow framework (i.e., data term plus smoothness term) significantly improves model performance. Both of these two points have been verified by our experiments, and we would like to supplement more results in the revised version.
>
> 3. Zero-Shot Resolution Generalization
>
> Thank you for your recognition of the robustness of our model under low-resolution input conditions. We also agree with your suggestion to test "zero-shot resolution generalization" as it would be valuable for further validating the operator properties and scale generalization capabilities of the model. However, we would like to clarify that PIVNO adopts a multi-scale input strategy during training. Specifically, the resolution of input image pairs is randomly sampled within a downsampling range from 1× to 4× (see Supplementary Materials B3). This training approach explicitly exposes the model to a continuous range of resolution distributions, enabling it to adapt to inputs of any scale. In other words, the scale generalization performance of PIVNO is learned progressively through this random scale training strategy. Therefore, the setting you suggested—“train the model on a single resolution (e.g., 128×128) and directly test on higher resolutions”—does not align with the core idea of our current training design. This setup would artificially limit the model's ability to model scale variations and would reduce its true generalization performance. Additionally, we believe that the "neural operator" property emphasized by PIVNO is more focused on its ability to map between image pairs (observation space) and fluid motion fields (function space), which differs from purely seeking scale invariance in images.
>
> Nevertheless, your suggestion has been highly insightful for us. We are currently conducting a set of "zero-shot resolution generalization" experiments, where the model is trained on a single resolution (128×128) and then evaluated at an unseen resolution (256×256). We expect to present the results of this test in the rebuttal in the coming days and further analyze its impact on the model's operator generalization ability. Once again, thank you for raising this thought-provoking question.
>
> 4. Inference Speed Comparison
>
> Thank you for pointing out the limitations of the inference speed evaluation. We would like to further clarify the statement emphasized in our paper, "avoiding the computational overhead of cost volume," which is intended to highlight the structural advantage in computational complexity of the RoPE-GA module compared to traditional methods that rely on explicit cost volume (CV) construction.
>
> Regarding your suggestion to directly compare the inference time with PIV-RAFT, we believe that, although both methods are comparable in terms of the task, the overall architecture differences (including feature extraction modules and the GRU update mechanism) make a direct runtime comparison potentially unfair and inaccurate in reflecting the actual contribution of the RoPE-GA module to the computational efficiency. Furthermore, it may not fully showcase the performance benefits brought by RoPE-GA.
>
> To more rigorously evaluate the efficiency advantage of our method's structural design, we have included a controlled ablation experiment in Supplementary Materials A3, where PIVNO-CV follows the RAFT-PIV architecture, employing traditional cost volume construction, while PIVNO represents our proposed method based on the RoPE-GA representation. The results show that the RoPE-GA method significantly reduces inference time while improving accuracy, validating the effectiveness and practical value of our "cost volume-free" design strategy.
>
> [1] Dual-frame Fluid Motion Estimation with Test-time Optimization and Zero-divergence Loss, in Neurips 2024.
>
> [2] Recurrent all-pairs field transforms for optical flow, in ECCV 2020.
>
> [3] Foundationstereo: Zero-shot stereo matching, in CVPR 2025.

---

> > ### Comment · Reviewer_PrER · 2025-08-04
> > **Reply to rebuttal**
> >
> > I appreciate the efforts the authors made during the rb period. I keep my original acceptance rating.

---

> > > ### Author Response · Authors · 2025-08-04
> > >
> > > I appreciate the efforts you made during the rebuttal period. Although you have already confirmed the final opinions, for the completeness of our work, please allow us to present our experimental results on Zero-Shot Resolution Generalization, as shown in the table below:
> > >
> > > | Method | Metric     | Uniform | Back-Step | Cylinder | JHTDB channel | DNS turbulence | SQG  |
> > > |--------|:----------:|:-------:|:---------:|:--------:|:-------------:|:--------------:|:----:|
> > > | Var    | 64²×1       | 3.66    | 1.15      | 2.11     | 1.62          | 4.05           | 2.59 |
> > > | Var    | 128²×1      | 3.45    | 0.78      | 0.84     | 1.37          | 3.00           | 2.03 |
> > > | Var    | 256²×1      | 3.33    | 0.87      | 0.57     | 1.36          | 2.88           | 2.09 |
> > > |  SR       | 64²×4       | 268.95  | 79.31     | 79.31    | 128.06        | 121.37         | 121.37 |
> > > | SR     | 64²×2       | 109.83  | 62.70     | 65.48    | 46.98         | 93.21          | 89.25 |
> > > |   SR      | 128²×2      | 108.37  | 59.47     | 49.50    | 58.55         | 98.28          | 92.11 |
> > > |   LI      | 64²×4       | 15.69   | 255.71    | 91.97    | 120.70        | 160.12         | 170.33 |
> > > | LI     | 64²×2       | 3.33    | 209.00    | 124.10   | 86.25         | 160.67         | 161.98 |
> > > |   LI      | 128²×2      | 2.33    | 326.01    | 98.12    | 109.08        | 165.97         | 181.58 |
> > > |   Std      | 64²×4       | 2.28    | 2.65      | 3.08     | 11.25         | 17.99          | 18.03 |
> > > | Std    | 64²×2       | 1.35    | 1.02      | 1.36     | 2.87          | 4.96           | 4.18 |
> > > |   Std     | 128²×2      | 1.36    | 0.91      | 1.10     | 2.81          | 4.70           | 4.08 |
> > > |   Std     | 256²×1      | 3.26    | 1.91      | 0.79     | 1.68          | 3.47           | 2.54 |
> > >
> > > Var represents the varying resolution, and SR represents super-resolution. Both are based on the training phase with an input resolution of 128² and output resolution of 128². Var refers to the output at different resolutions during the testing phase, while SR represents outputs at different magnifications of super-resolution. For instance, 64²×4 means that the input resolution during the testing phase is 64², and ×4 indicates a 4x super-resolution output, resulting in an output resolution of 256². LI refers to using linear interpolation, while Std represents the standard PIVNO setup in our paper.
> > >
> > > As shown by Var, the PIVNO model performs well even when faced with unseen sampling rates (64²×1, 256²×1) during the testing phase. This is in line with our expectations for the PIVNO operator, proving that PIVNO learns a general mapping from particle images to fluid motion fields, demonstrating its generalization capability with respect to resolution.
> > >
> > > From SR, we observe that the model performs worse in super-resolution scenarios (64²×2, 64²×4, 128²×2). The reason for this is that the training phase did not involve super-resolution scenarios, making it unrealistic to rely solely on the model's structure for super-resolution capabilities. This also indirectly verifies the validity of our super-resolution training strategy.
> > >
> > > Finally, the LI and Std settings are provided for comparison, demonstrating that our super-resolution performance far exceeds that of LI. Interestingly, the Uniform term for LI performs relatively well because Uniform represents a uniform flow, where the velocity field is consistent throughout. This characteristic aligns well with the requirements of linear interpolation, as linear interpolation assumes a smooth and uniform change between points, making it suitable for estimating uniform flows.

---

### Official Review · Reviewer_7Y2L · 2025-07-03

**Clarity:** 3
**Significance:** 3
**Originality:** 2
**Rating:** 4
**Confidence:** 3

**Summary:**

The paper presents a new method for Particle Image Velocimetry (PIV) using a Neural Operator framework (PIVNO).  The framework leverages a position-informed Galerkin-style attention operator to capture global flow structures while supporting resolution-adaptive inference. It incorporates self-supervised fine-tuning based on physical divergence constraints, helping the model generalize from synthetic to real-world experiments without requiring labeled data.

**Questions:**

How does PIVNO perform when applied to completely novel fluid dynamics conditions that differ significantly from the training scenarios? Are there specific flow regimes where the model’s performance might degrade.

**Ethical Concerns:**

["NO or VERY MINOR ethics concerns only"]

**Final Justification:**

The author has addressed my concerns and I want to maintain the score.

**Limitations:**

The author discussed the relevant limitations.

**Paper Formatting Concerns:**

No Formatting Concerns

**Quality:**

3

**Strengths And Weaknesses:**

**Strengths**

1. One of the key strengths of PIVNO is its ability to perform resolution-adaptive inference. The model can adapt to varying input resolutions and still provide accurate high-resolution flow field outputs. This makes it highly versatile in real-world applications where the region of interest may have different resolution requirements.

2. The self-supervised fine-tuning mechanism based on physical divergence constraints enables the model to adapt from synthetic datasets to real-world conditions without requiring labeled real data.

**Weaknesses**
1. The self-supervised fine-tuning allows PIVNO might not be fully reliable for all real-world datasets. The fine-tuning mechanism still relies on certain assumptions, such as the physical divergence-free condition, which may not always hold or may be difficult to enforce in some experimental setups.

2. The current PIVNO framework is designed for 2D flow field estimation. This limitation restricts its ability to fully capture the three-dimensional (3D) nature of many fluid dynamics problems. Extending the model to handle 3D flow fields is a critical next step to broaden its applicability to more complex and realistic fluid systems.

3. Real experimental conditions can vary widely in terms of lighting, particle distribution, and other environmental factors, which may not always be fully covered by the synthetic training datasets.

---

> ### Author Rebuttal · Authors · 2025-07-30
>
> Thank you very much for your valuable feedback and suggestions! We highly appreciate your comments and treat them carefully. For each issue raised, we will make modifications or clarifications as needed. Below, we briefly discuss the points raised:
>
> 1.Limitations of 2D Estimation
>
> We fully agree with your view that "extending to 3D flow fields is a crucial next step to improve the model’s applicability to more complex and realistic fluid systems." Our focus on 2D in this work is based on a careful assessment of existing mainstream 3D methods. Current 3D flow estimation methods mainly adopt particle tracking velocimetry (PTV), which requires a prerequisite  particle detection phase followed by the paticle matching and flow estimation process. This approach is highly sensitive to detection quality and prone to error propagation, and produces sparse Lagrangian motion fields that are difficult to use for dense Eulerian field reconstruction. In contrast, PIVNO follows the PIV paradigm, estimating dense Eulerian flow fields end-to-end directly from particle image pairs, thus avoiding the instability and information loss introduced by intermediate steps. Therefore, we first establish a solid and reproducible baseline in 2D scenarios by integrating physical priors. On this foundation, we have already started developing  tomography-based 3D PIVNO to further extend applicability to complex real-world 3D flows.
>
> 2.Physical Assumption Limitations of the Self-Supervised Mechanism
>
> As you pointed out, we indeed introduce a divergence-free constraint based on the incompressible flow assumption, which may not fully hold in some experimental setups or real applications. It is important to emphasize that the divergence-free constraint is neither inherent nor the only physical constraint usable in the PIVNO framework. It serves as a physical prior tailored to the incompressible flow characteristics of our current datasets. For other fluid types, the PIVNO framework can similarly replace or extend physical constraints to fit new assumptions or scenarios. This constraint is flexible and should be set by the researchers according to the physical context and prior knowledge of the problem.
>
> 3.Generalization and Performance Degradation
>
> We fully recognize that the complexity of fluid dynamics makes it impossible for the model to maintain ideal performance under all conditions. Hence, we propose a self-supervised fine-tuning mechanism specifically to address potential performance degradation in real and complex conditions. In our experiments, the three selected real scenarios differ significantly from the synthetic training data in flow states, particle density, illumination conditions, etc. As shown in Table 3 of the main text, the model’s performance does degrade noticeably when encountering novel flow states significantly different from training. However, self-supervised fine-tuning effectively mitigates this degradation and substantially improves the final estimation accuracy, enabling the model to remain robust and adaptable in complex real-world scenarios.

---

> > ### Comment · Reviewer_7Y2L · 2025-08-06
> >
> > The author has addressed my concerns and I want to maintain the score, which is already positive.

---

> > > ### Author Response · Authors · 2025-08-07
> > > **Thank you**
> > >
> > > Thank you so much for taking the time to read our paper and the responses.

---

### Official Review · Reviewer_gQBY · 2025-07-05

**Clarity:** 3
**Significance:** 3
**Originality:** 3
**Rating:** 5
**Confidence:** 3

**Summary:**

This work proposes PIVNO, a neural method that directly estimates the mapping from paired particle images to flow fields. It addresses three key challenges:
- introducing the PIV Neural Operator to ensure that the estimated mapping lies within a physically feasible solution space;
- proposing a self-supervised fine-tuning scheme to bridge the sim-to-real gap;
- incorporating a super-resolution module to produce high-resolution flow estimations.

Experiments are conducted on three synthetic datasets and three real-world datasets, all of which demonstrate the effectiveness of the proposed approach.

**Questions:**

1. After fine-tuning on the real dataset, could you also evaluate the fine-tuned model on the synthetic dataset? This analysis would help assess whether the fine-tuning leads to overfitting on real data and a loss of generalization to synthetic scenarios.

2. It would be valuable to involve domain experts (e.g., fluid mechanics researchers) to qualitatively evaluate the correctness of the estimated flow field.

3. In addition to reporting the average endpoint error, could you also provide statistics such as the mean and standard deviation of the error distribution? This would offer a more comprehensive view of the model’s performance and robustness.

**Ethical Concerns:**

["NO or VERY MINOR ethics concerns only"]

**Final Justification:**

All my concerns have been addressed. This is an interesting work. I maintain my score as 'Accept'.

**Limitations:**

yes

**Quality:**

3

**Strengths And Weaknesses:**

Strengths:

1. It is interesting to bypass the traditional cost-volume construction and directly estimate the flow field. The idea of the Galerkin-type attention operator is novel, and the experimental results support its effectiveness.

2. The proposed framework is both sound and innovative.

3. The experiments are comprehensive, evaluating the method on both synthetic and real-world datasets.

Weaknesses:

1. The organization of the experimental section is somewhat confusing. For example, some ablation studies are already presented in Section 4.1 (e.g., Table 3), yet a separate section titled "Ablation Study" is also included. While I appreciate the thoroughness, I suggest reorganizing the experimental section for better clarity and coherence.

2. I have concerns regarding the soundness of the evaluation metric used—Average Endpoint Error (AEE). Have the authors considered reporting the maximum error or the per-timestep error? This could help assess error accumulation over time.

---

> ### Author Rebuttal · Authors · 2025-07-30
>
> Thank you very much for your valuable feedback and suggestions! We greatly appreciate your comments and treat them carefully. For each issue raised, we will make modifications or clarifications as needed. Below, we provide a brief discussion of the concerns raised in the review:
>
> 1.Unclear Experimental Structure
>
> You pointed out that the structure of the experimental section is somewhat confusing, especially the redundancy between Section 4.1 and the separate "Ablation Study" section. We fully understand and acknowledge this. The reason for presenting Table 3 separately is that it not only includes ablation experiments on the self-supervised loss components but also summarizes the overall performance of the self-supervised strategy on both synthetic and real datasets. Therefore, we believe it serves as a key indicator of the self-supervised strategy's effectiveness and deserves a separate presentation. The "Ablation Study" in Section 4.3 focuses on ablation analysis under other experimental settings. However, we realize that the current structure may not effectively guide readers through the hierarchy of experiments. To address this, we will include additional clarifications in the revised version to explicitly outline the logical structure of the experimental section, thereby improving clarity and coherence.
>
> 2.Rationality of Evaluation Metrics
>
> Currently, in the PIV field, the Average Endpoint Error (AEE) is the mainstream evaluation metric. Moreover, many related works have not released their source code, which makes it less possible to conduct comparisons using alternative metrics. We fully agree with your concern about the rationality of evaluation metrics. To address this, we have additionally reported the maximum error（Max）and standard deviation (Std) of the error distribution on Synthetic Dataset 1. The experimental results are as follows (unit: pixels):
>
> | Metric | Method   | Uniform | Back-Step | Cylinder | JHTDB channel | DNS turbulence | SQG |
> |:------:|:--------:|:-------:|:---------:|:--------:|:-------------:|:--------------:|:---:|
> |  Max   | 64²×4    |  0.662  |   0.660   |  0.745   |     2.653     |      2.564     |2.562|
> |  Max   | 64²×2    |  0.228  |   0.318   |  0.309   |     2.060     |      1.694     |1.491|
> |  Max   | 128²×2   |  0.303  |   0.418   |  0.370   |     1.843     |      1.185     |1.165|
> |  Max   | 256²×1   |  0.660  |   3.457   |  2.328   |     9.604     |      8.995     |2.706|
> |  Std   | 64²×4    |  0.005  |   0.008   |  0.008   |     0.031     |      0.046     |0.027|
> |  Std   | 64²×2    |  0.003  |   0.004   |  0.004   |     0.009     |      0.019     |0.011|
> |  Std   | 128²×2   |  0.003  |   0.003   |  0.004   |     0.009     |      0.016     |0.007|
> |  Std   | 256²×1   |  0.010  |   0.029   |  0.007   |     0.029     |      0.057     |0.011|
>
> It is particularly noteworthy that under the 256²×1 resolution setting, the model exhibits significantly higher maximum error and standard deviation compared to other settings. Upon further investigation, we found that this is caused by a few extreme error points near the boundary regions, which raise the overall statistics. In contrast, the ×4 and ×2 resolution settings inherently introduce spatial averaging (filtering) effects during the super-resolution reconstruction process, which smoothes out the local noise and thereby reduces both the maximum error and the error variance.
>
> 3.Evaluation of Fine-Tuned Models on Synthetic Datasets
>
> Our method primarily focuses on training the model on synthetic datasets to learn a general mapping from input particle images to output motion fields, followed by self-supervised fine-tuning in specific real-world scenarios to enhance performance for the task at hand. As our main goal is to improve performance in these specific real-world tasks, the generalization of the fine-tuned model back to synthetic datasets was not our primary consideration. However, we understand the reviewers’ concern about the generalization ability after fine-tuning and the consistance on its importance. Therefore, we have conducted experiments to evaluate the generalization of the fine-tuned model on the synthetic dataset. The results are as follows (unit: pixels):
>
> | Resolution | Uniform | Back-Step | Cylinder | JHTDB channel | DNS turbulence | SQG   |
> |:----------:|:-------:|:---------:|:--------:|:-------------:|:--------------:|:-----:|
> | 64²×4      | 4.3100  | 2.1492    | 2.3437   | 1.1926        | 1.1089         | 0.9776 |
> | 64²×2      | 6.8775  | 3.5106    | 2.5500   | 1.8067        | 1.0801         | 1.0307 |
> | 128²×2     | 6.9424  | 3.0903    | 3.0121   | 1.8427        | 0.9936         | 1.0770 |
> | 256²×1     | 7.4281  | 3.2023    | 3.7374   | 2.6118        | 2.2822         | 2.4208 |
>
> Experimental results show that after fine-tuning on real-world datasets, the performance of PIVNO on synthetic datasets exhibits  a noticeable decline. This reflects the nature of fine-tuning as a process of adapting a general model to specific scenarios. Given that PIVNO contains only 2.4M parameters, expecting it to maintain strong generalization across both synthetic and real domains without any trade-offs is somewhat overly demanding. Afterall, the core value of our model lies in its adaptability—achieving effective performance in diverse real-world situations through lightweight fine-tuning—rather than aiming for a one-size-fits-all solution through a single training process.
>
> 4.Involvement of Domain Experts
>
> We fully agree with your point that qualitative assessment of flow field estimations by domain experts (e.g., fluid mechanics researchers) would be highly valuable. The datasets and baselines used in our study are widely recognized and scientifically grounded in the field, effectively reflecting the accuracy and applicability of our method. To promote interdisciplinary collaboration and further validate the physical reliability of the model, we will open-source our code and sincerely welcome experts in fluid mechanics and related fields to participate in qualitative evaluations and assesments, jointly advancing the application and development of our method to real-world flow problems.

---

> > ### Comment · Reviewer_gQBY · 2025-08-06
> > **Response to rebuttal**
> >
> > Thanks for your response. All my concerns have been addressed. I maintain my score as 'Accept'.

---

> > > ### Author Response · Authors · 2025-08-06
> > > **Thank you**
> > >
> > > Thank you so much for taking the time to read our paper and the responses.

---

### Decision · Program_Chairs · 2025-09-17

**Decision:**

Accept (poster)

**Comment:**

This paper proposes PIVNO, a neural operator approach for Particle Image Velocimetry that directly maps paired particle images to velocity fields. The framework combines a position-informed Galerkin-style attention operator (RoPE-GA) for global flow capture, a Conv-GRU module for iterative refinement, and a self-supervised fine-tuning scheme grounded in physical divergence constraints. PIVNO supports resolution-adaptive inference and incorporates a super-resolution module. Experiments on three synthetic and three real-world datasets demonstrate strong performance compared to multiple baselines.

Reviewers highlighted the paper’s novelty, well-motivated architecture, and comprehensive evaluation. PIVNO effectively bypasses traditional cost-volume approaches, adapts to real-world conditions without labeled data, and shows clear improvements in both accuracy and versatility. Weaknesses include some issues in the experimental section, limited 2D scope, incomplete ablation of certain loss terms, missing inference speed comparisons, and the need for further justification of some architectural choices, such as Conv-GRU refinement.

During rebuttal, the authors clarified the network design, committed to reorganizing experiments for clarity, and agreed to include additional metrics and comparisons in the final version. Based on the novelty, strong empirical results, and satisfactory rebuttal, I recommend ACCEPT (Consensus Reached). The authors should implement the promised improvements in the final version to fully address reviewer concerns.